# Activation of Investigator-Initiated Clinical Trials with a Pharmaceutical for Cancer Patients before and after Post-Millennial Changes of Regulations in Germany and Europe

**DOI:** 10.3390/cancers14051308

**Published:** 2022-03-03

**Authors:** Wolfgang E. Berdel

**Affiliations:** Department of Medicine A (Hematology, Hemostaseology, Oncology, Pneumology), University Hospital of Muenster, 48149 Muenster, Germany; berdel@uni-muenster.de; Tel.: +49-251-835-2672

**Keywords:** Good Clinical Practice Guideline (GCP), Clinical Trials Directive (CTD), university hospital structure and finance system in Germany, investigator-initiated trials (IIT), financial burden for clinical trials

## Abstract

**Simple Summary:**

This opinion paper describes the regulatory hurdles for a clinical oncologist and physician scientist to activate an Investigator-Initiated Trial (IIT) before and after 2004 with German regulation as an example. Changes in legal framework with impacts on time and costs to activate a clinical trial are described. Evidence needed to reach the objective of higher patient safety and trial quality by European Union (EU) Clinical Trial Directive (CTD) 2001/20 is discussed.

**Abstract:**

Shortly after the beginning of the year 2000, multiple legal changes with impacts on the regulatory framework of clinical trials became effective almost simultaneously. They included the European Union (EU) Clinical Trial Directive (CTD) 2001/20 followed by major changes in national drug laws, the change in the legal status of German University Hospitals (1998), and a new disease-related groups (DRG)-based reimbursement system for hospitals in Germany (2000). Together, these changes created enormous bureaucratic and financial inhibition of activation and conduct of academic investigator-initiated clinical trials (IIT). Examples for activating clinical trials in oncology before and after 2004 are outlined and discussed, focussing on extended time frames, the establishment of centralized responsibility structures and the exploding financial consequences. In addition, the evolution of trial numbers and the distribution of trial initiators between “commercial” and “academic” over time are discussed together with the occurrence of clinical registries. At the same time, progress in molecular biology led to a plethora of new targets for effective pharmacological therapy of life-threatening diseases such as cancer, and the overall number of clinical trials has not decreased. Yet, judging the regulatory and administrative hurdles between scientific study design and first-patient on trial before and after 2004 and weighing these against the lack of evidence that this regulation has achieved its goal to enhance patient safety and trial quality, the necessity to completely overhaul this CTD becomes obvious. A main goal of such an initiative should be to minimize bureaucracy. For the specific situation in Germany, relocation of responsibility and freedom to operate in University Hospitals and Medical Faculties back to the physician–scientists and reduction in interference by legal divisions should be a goal as well as increasing the public financial support for IITs.

## 1. Trial Activation and Conduct before 2004

In early 1981, I wrote a study protocol for a *monocentre phase I trial* studying an “investigational medicinal product” (IMP) of a novel class of antitumor compounds with a new mode of action. This protocol was based on approximately 7 years of preclinical scientific work in laboratories of a Max Planck Institute and some university laboratories, summarized in several scientific publications on the pharmacodynamics and toxicology results obtained with the ether lipid IMP class (later reviewed in [1]). As necessary at this time, the competent authority in Germany (Federal Health Office, Bundesgesundheitsamt, BGA) was informed of this trial project by a letter along with two publication reprints and a folder containing the study protocol, a chemical and microbiological examination report of the IMP, a brief document on safety and toxicology observations in animals and a patient insurance confirmation. The BGA responded to the package within a few days issuing a trial number and an additional statement that the regional state authorities have been informed simultaneously on the trial project. In parallel, the University Ethical Board held a hearing on the trial with the investigators present and upon consultation gave the “green light”. The trial was started within 1 month from submission to BGA and the Ethical Board. Trial conduct was strictly according to the Declaration of Helsinki in its actual version and Federal Drug Law (Arzneimittelgesetz, AMG). The calculated costs for this trial was Deutsche Mark (DM) 60,000 (later exchange rate DM:Euro was 2:1), essentially caused by the IMP synthesis and a part-time special study-physician for clinical surveillance of the 16 patients. First results were internationally published in a peer-reviewed journal within 2 years [2]. The IMP studied did not make it to approval due to a small therapeutic window, but a successor within the same class of compounds later obtained approval [3]. 

At the same time, I was co-investigator in several clinical *multicentre studies of phases II and III* initiated by academic oncologist investigators. Among these was one of the first phase II trials in Germany on the activity of a cisplatin doublet in non-small-cell lung cancer (NSCLC). Activation and conduct of those trials followed the same pathway. Multicentre cooperation between clinical investigators and a study statistician was discussed and agreed upon during a personal study meeting, legal information on the BGA and the regional authorities was performed by the leading investigator, documentation of study results was performed on detailed case report forms developed by the investigators for this trial, and statistics were performed by the statistician in the leading trial centre. The calculated costs for such a trial were negligible as treatment was reimbursed by health insurances as a standard of care and patient surveillance plus documentation were performed by the investigators as standard tasks within clinical patient care. Again, publication of this trial with results on 26 patients appeared within 1 year in an international peer-reviewed journal [4]. The treatment of advanced NSCLC with cisplatin doublets soon became standard of care.

The three-to-four decades before 2001 in clinical oncology research were characterized by large, academic, multicentre “*therapy optimization studies*” (TOS), with the objective to improve the algorithms of often multidisciplinary cancer treatment by randomized comparisons step-by-step. These studies were open for major oncological and haematological indications, and I—as all other haematologists and oncologists in training at this time—reported and treated the majority of my patients within these non-commercial TOS. TOS were activated and conducted as described above with few central regulatory hurdles within the responsibility of the investigators. Indication-specific investigator-initiated *study groups* were founded and were essential for trial progress. Trial conduct was strictly according to the Declaration of Helsinki and Federal Drug Law. Some TOS were initially financed by the German Federal Ministry of Education and Research. Later, financial support of these studies was provided by non-profit charity organisations, the German Cancer Aid (Deutsche Krebshilfe, DKH) or for paediatric oncology also by Deutsche Kinderkrebsstiftung (DKS), upon independent review of scientific applications. Budgets for such a trial were variable, but with exceptions they were well below DM 500,000. Standard of care for major malignant disease entities was defined by the knowledge based on these trials. 

In paediatric oncology, the step-wise optimization of multimodal therapy by TOS had a profound impact on cancer cure rates in children and adolescents across many cancer entities [5]. Success in adult haematology/oncology was slower, but could clearly be observed in acute leukaemias, lymphomas including Hodgkin’s disease and several solid tumours. Examples are the subsequent study generations of the German Acute Myeloid Leukemia Study Group (AMLCG) which improved the outcome of patients with acute myeloid leukaemia (AML) by modification of chemotherapy and supportive care (Figure 1). Since the 1990s, all study groups of the German Cancer Society (DKG) and its subgroups have worked according to Standard Operating Procedures (SOP) edited and agreed upon by investigators from all study sites [6]. Such a trial structure in oncology facilitated continuous therapeutic improvement by scientific consensus on treatment changes within a general study outline. Parallel to these cooperative study groups, associated research helped to better understand disease biology, e.g., disease prognostic subgroups, genetic drivers and microenvironment interactions, and identified new targets for specific therapeutic intervention.

## 2. Changes in the Legislation around 2000

Three specific events around the change of millennium emerged into a completely different scenery for clinical trials in haematology, oncology and also other medical fields, with dramatic consequences.

In Germany, *university hospitals* (*UH*) *had to adopt a new legal status*. Before 1998, UH were non-independent operations of the German Federal States. In particular, they had no legal independence and relied on small administrations, which simultaneously operated for the corresponding medical faculties of the universities. With the new legal status, most university hospitals became corporations under public law [8] which induced a dualism between the medical faculty and university hospital with areas of conflict and legal independence of the university hospital, resulting in considerable workforce expansion in the legal divisions and changes in responsibilities. Physician–scientists were no longer able to activate an IIT in their own responsibility but depended on complicated decision pathways by the legal department of the hospital with time and cost consequences, as outlined below.In 2003, the legal basis of *hospital financing* was switched to a German adaption of the Australian *Disease-Related Group* (*G-DRG*) *system*. The aim of G-DRG was to limit the “cost explosion” in the health system. It is not the topic of this opinion paper to judge whether such a “cost explosion” ever existed or whether this objective was reached. However, today roughly two-thirds of German UH are in deep financial deficit and the annual deficit accumulated to EUR >500 million in 2020 [9]. At the same time, the dual system of financing for the UH, with insurance reimbursement for standard of care and with state grants for research and teaching activities, are by no means sufficient to cover the dramatic cost increases for clinical IITs, and an adequate build-up of public grants or grants by charity organizations for the conduct of clinical trials, in particular IITs, is lacking behind. Thus, today many IITs are financed or co-financed by the pharmaceutical industry, which changes the meaning of the term IIT.*Good Clinical Practice* (*GCP*) *Guidelines* were issued by the European Union as Clinical Trials Directive 2001 (EU CTD 2001/20) throughout Europe, which was followed by national changes in Drug Laws (2004, in Germany 12. Novelle, AMG). Whereas the two structural and legal changes mentioned above are more or less Germany-specific, the GCP-guidelines have an impact in Europe and the US. This legal framework, implemented with the objective to be a quality standard for the design, conduct, performance, monitoring, auditing, recording, analysis and reporting of clinical trials, and hereby to improve trial quality and patient safety, has caused tremendous difficulties and delays in the activation and conduct of clinical trials. An important example of the new bureaucratic hurdles is the institution of a “sponsor”. A sponsor is responsible for initiation, management and/or financing of a clinical trial, and its institution has transferred the legal responsibility for a trial from the investigators and clinicians to a central legal institution (or individual). The EU CTD 2001/20 has resulted in a cost explosion in the performance of academic clinical trials designed to develop optimized standards of care, which have to comply with the same regulatory and administrative requirements as commercial drug development through clinical phases I to III.

Specific consequences of these three changes “cooperating” in the sense of *emergence* for activating (and conducting) IITs are exemplified below. *Emergence* occurs when a system develops properties which its single parts alone do not have, i.e., properties or behaviours which emerge only when the parts interact.

## 3. Trial Activation after 2004

A physician–scientist who wants to activate a clinical *monocentre phase I trial* (IIT) under the modified legal requirements described above is obliged to complete the following path: Composing a scientific study protocol including relevant preclinical studies and publications is not sufficient anymore. Documents summarizing the Investigational Medicinal Product (IMP) characteristics within a dossier (IMPD) and the major non-clinical research results of pharmacodynamic, pharmacokinetic, safety and toxicology studies summarized in an Investigator “Brochure” (IB) have to be prepared according to multiple and specific guidelines by the International Conference for Harmonization of Technical Requirements for Pharmaceuticals for Human Use (ICH) adopted by European Medicines Agency (EMA) and US Food and Drug Administration (FDA). Next, a sponsor has to be found either in the commercial private sector, in which multiple Clinical Research Organizations (CRO) have started to offer sponsor services, or in the academic sector. Here, a university or a UH is the sponsor and Coordinating Centres for Clinical Studies (Koordinierungszentren für Klinische Studien, KKS) or Centres for Clinical Studies (Zentrum für Klinische Studien, ZKS) were founded and have been delegated tasks. Work steps provided by CROs and KKS/ZKS are defined by multiple standard operating procedures (SOP) which are not harmonized and constantly subject of change. As an example, in 2012 a quotation of a KKS/ZKS to contribute to (not to completely take over) sponsor tasks for a Phase I single-centre IIT involving 20 patients, including document checking, insurance, notification and communication with the ethical board, competent federal authority, and regional governments, data management, pharmacovigilance and reporting (AE/SAE/SUSAR) with MedDRA coding, yearly safety reports (DSUR), monitoring, auditing and help with government inspections, amounted to EUR 160,000. Written contracts have to be negotiated and finalized for this task with the legal department of the faculty/UH. In addition, the legal departments request additional written contracts with central departments of the UH, such as pharmacy, clinical chemistry and radiology, covering the costs for the study-specific investigations for the IIT. The requirements for a specific patient insurance have not changed. An independent Data Monitoring Committee (DMC) with charter and contracts has to be instituted according to ICH guidelines. Who can cover these costs? One option is a pharmaceutical company with interest in the study results, which changed the meaning of IIT since companies often claim involvement in the study design as a condition. Support by public or charity institutions such as the DKH can be applied for but the budget of both public funding and donation-based charities in the view of the high cost of each individual trial limits the number of academic studies that can be performed nationwide. Application and review timelines, then contract work between support institution and UH legal department, often exceed one year. Subsequently, the sponsor works out a detailed responsibility distribution table for the conduct of the study. Study content has to be uploaded into public data bases (EudraCT and/or ClinicalTrials.gov) using old-fashioned and not very user-friendly software. Then, approvals by the federal competent authority and the ethical board, bound by legal timelines, have to be applied for, and upon statement of formal correctness (10-day limit), follow within 30 days (60 days for multi-institutional trials) with costs of EUR >3000. A typical file submitted by the author to the Paul-Ehrlich-Institute (PEI) in 2016 to be validated within this timeline contained documents exceeding 3000 pages. In case of a question-and-answer round, another 10–20 days are needed. Further, the regional state authorities have to be informed, with invoices returned for this information. Only then are specific initiation meetings to be conducted, with all persons involved in study conduct present and signing further paper work, and finally the sponsor gives the “green light”, a written process signed by deans or UH directors which alone can take >1 week. Only a few of these processes can be managed in parallel; the main procedure is characterized by “one step after the other”. So, in summary, initiation of a monocentre phase I clinical IIT in Germany costs approximately EUR 200,000 for legal and administrative costs and has a timeline of well over 1.5 years, if everything runs smoothly. 

In case a group of physician–scientists and clinical investigators plans to start a *multicentre IIT* after 2004, including the above-described optimization trials (TOS) of multimodal treatment paths based on already approved agents, what hurdles have to be undertaken in addition to those described above for a monocentre study? Agreement on the trial design within a scientific meeting of the investigator-initiated study group and the writing of a study protocol is still the starting point of all IITs, but the formal and legal responsibility for such a venture also has been completely transferred from the clinicians to central, or in an even more complicated manner, decentralized legal departments and institutions. As a specific aspect, the trial site contracts (Prüfzentrumsverträge) have to be mentioned. UH legal departments quote the European Law on State Aid (Europäisches Beihilferecht) to request detailed calculations of the full study costs (Vollkostenkalkulation) to be granted to the study centres including faculty “overheads” and incorporation of this into detailed contracts of cooperation between the coordinating sponsor centre and the participating centres. This is in contrast to the statement of the EU commission “that it certainly does not wish to interfere with cooperation partners’ freedom to make contractual arrangements even when contract research is involved. The Commission’s recommendations should be a help, but not become a straitjacket” [10] and completely neglects the scientific interests of clinician scientists to cooperate in innovative clinical trials with the aim to drive progress and publish together. Then, detailed contracts of cooperation are exchanged, which needs 4–6 rounds of mutual corrections between the legal departments of the participating UHs addressing financial agreements, but also includes details such as whether the project legally is named a “clinical study” or “clinical examination”. In addition, by-contracts are written, which distribute rights and tasks concerning the European General Data Protection Regulation 2016/679 (EU-GDPR) that became direct law in the member states in 2020. Again, multiple rounds of mutual corrections are involved as harmonization of the interpretation of GDPR and oversight performance by the different data protection officers is largely lacking. Contract negotiations between centres can easily last longer than 6 months. The procedures concerning ethical board and competent authority decisions foresee timelines of 60 days. Directly following in this scenario, organization of initiation meetings with all participating centres and obtaining the “green light” can well require another 2–3 months. For a phase III IIT involving 10 centres, the sponsor costs alone can amount to approximately EUR 500,000 and the initiating investigators easily count several hundred e-mails or letters necessary to move the regulatory initiation process forward “step-by-step”. Time frames—and this only refers to personal experience as comparative data are not easily found—from writing and agreeing on an IIT study protocol to the “green light” for all centres can be easily >2 years. In contrast, the fastest activation for an industry-sponsored multicentre study in the first centres with a commercial CRO as sponsor, as I can remember, was 6 months. This, and the fact that commercial CROs sometimes offer financially more favourable quotations, documents the problems that the public system has to compete with.

## 4. Emergence and Consequences of Change in UH Legal Status, G-DRG, and EU-CTD 2001/20 for the Activation of an IIT after 2004

The complicated and bureaucratic procedures described illustrate the impact of all three system changes after 2004 for IIT activation. The UH legal department has taken over responsibility for all formal and legal aspects of the trial, as a sponsor, the UH or university delegate tasks according to AMG and GCP to a KKS/ZKS or unwillingly accept a commercial sponsor for an academic trial. The sponsor then helps to execute the multiple bureaucratic tasks provided by AMG and GCP inside the UH and with all regulatory institutions by a “one step after the other” approach. The clinical investigators are at the receiving end. They communicate with the various partners involved by hundreds of messages to confirm compliance with the formal regulations, to speed up the lengthy process, to find solutions for various financing and regulatory issues, and wait and wait and wait.

In my experience, all individuals acting in this regulatory process act with friendliness and diligence, but also often enormous hesitation to accept the responsibility. They justify their actions and also their hesitation to accept responsibility with their position as guardians of the legal system, and this is correct. Only few of them ever have been involved in the conduct of a clinical study or in direct patient care. Thus, these persons should not be targeted when frustration is high, but instead, the European and national legislators should be called upon for reformation of the law in the interest of medical progress. In stark contrast to the scientific and medical area or in social sciences, such major changes in the legal framework as the three described above are rolled out without any comparative field study to ask whether they reach their objectives or to learn about advantages and disadvantages. A concluding statement within the legislation parliament proposals often reads: Alternative—None. The way these regulations are built is completely devoid of evidence.

A literature search (PubMed, multiple search terms) for evidence that GCP reaches its aim to improve the quality of clinical trials and increase safety for patients within trials in comparison to the situation before 2004 remains without success. Although this argument is always promoted by the regulatory institutions involved, it is not based on scientific evidence. On the contrary, multiple publications have discussed the weaknesses and problems connected with over-regulation of academic studies, of which only few concerned with oncology are quoted here [11,12,13,14,15,16,17,18]. Editorials of large international meeting reports signed by multiple clinical investigators (Am. Soc. Hematol. (ASH) 2017) have shed light on the absurdity of some aspects in the work of sponsors under GCP [19]. Some journals have even devoted complete Special Issues on this field [20] and scientific societies such as the German Society for Hematology and Oncology (DGHO) have published a multitude of statements on the impact of details of new trial regulations and have pointed out that not all trials need the same degree of regulation [21]. Publications on how “bureaucracy is strangling clinical research” have been signed by more than 1000 senior European researchers in haematology alone [22], and the lack of scientific evidence supporting any beneficial effect of CTD 2001/20 has been stated before [23,24].

All these publications of the scientific community were without much impact in the legislative and regulatory world. However, in 2013 the European Commission published a proposal for a new Clinical Trial Regulation (CTR) motivated by the argument that the number of clinical trials had decreased by 25% between 2007 and 2011 and that the costs for clinical trials and the time to activate a clinical trial had increased [25]. This was translated into the CTR 536/2014, which after 7 years of implementation difficulties, in part due to the complicated Clinical Trial Information System (CTIS) portal, will take effect in early 2022. This CTR was met with hopes of lower bureaucratic hurdles, but it was also already criticized before implementation to neither fully address nor solve the key problems also outlined by the EU proposal [26,27]. 

Not only the EU directives have caused these hurdles on the way to initiating an academic trial. The specific consequences in Germany were induced by all three interacting legal changes described, which in the sense of emergence led into a qualitative system change. The system changes induced are harmful to medical progress as they distract investigators from the scientific and clinical heart of the matter and as they divert millions of EUR of scarce funds for clinical research from public and philanthropic sources into a huge bureaucratic industry, claiming high objectives but without any scientific evidence to reach these. Further, higher numbers than ever of specialized, well-educated persons work in the regulatory rear and not at the front of patient care or clinical science, which adds to the rising problem of staff shortage in clinical research. 

With this said, *resilience of the clinical investigators and the trial system* on the other hand is remarkable. The EU proposal [25] stated a decline in trial numbers by 25%, but yearly trial statistics of both German Competent Authorities (Bundesinstitut für Arzneimittel und Medizinprodukte, BfArM and Paul-Ehrlich-Institut, PEI) cannot really substantiate this. Whereas a small decline in trial applications to the BfArM can be seen in the published statistics [28], the numbers of trial applications to the PEI are increasing, an observation which reflects the enormously increasing number of potential pharmacological targets for biologicals such as monoclonal antibodies [29]. However, the vast majority of these trials today has a commercial basis and the *distribution of study types between* “*commercial*” *and* “*academic*” is a concern. As examples, the number of clinical AMG-conform TOS performed by the German–Austrian–Swiss Society for Pediatric Oncology and Hematology (GPOH) dropped from 33 in 2002 to 2 in 2017 [30]. At the same time, so-called “*clinical registries*” increased from 1 to 28 [30]. In the ethics committee that reviews oncology trials in Helsinki, academic drug trials decreased from 20 to 5 between 2003 and 2005 [31]. A spot check of clinical trial applications at the ethical board of the physicians chamber of Westphalia-Lippe and the Westphalian Wilhelms University of Muenster revealed a similar trend, with decreasing AMG-conform IITs from 6 in 2006 to 2 in 2012, with almost constant numbers of commercially induced AMG trials (38 in 2006 and 29 in 2012) and an increase in clinical registries not bound to follow GCP-regulation from 1 to 11 over the same period. Similar observations were reported in a detailed survey on the impact of EU legislation on clinical research in Europe [32]. Clinical registries can help to maintain the quality of clinical care by providing centralized data collection and reference evaluation while circumventing the increased costs of clinical trials, but they cannot induce innovation into standard treatment algorithms. Further, to cover the costs of a GCP-conform IIT, a pharmaceutical company with interest in the study results often partially finances the studies, which changes the meaning of IIT in case the company insists on being involved into the study design and thereby limits the freedom of the academic investigator to design the study based on medical need and knowledge alone. Investigators often wonder why GCP-caused trial regulation was not designed in an evidence-based fashion, why it was never published after peer-review and never completely updated after scientifically assessing beneficial or detrimental consequences [23,24], and whether the shift from “academic”- to “pharmaceutical industry”-initiated studies was indeed an important driver for the 2001 regulations. However, in order to not introduce an unfair balance between industry-initiated studies and IITs, particularly during recent decades, therapeutic progress in oncology has also been enormously accelerated by specifically addressing new targets and pathways in the molecular pathophysiology of cancer cells, their microenvironment and immunology. This was and is often achieved by close cooperation between the pharmaceutical industry and academia on the laboratory and clinical level. 

Last but not least, an *International Drift* for clinical trials and trial sites, which would need GCP conformity from North America and Europe to developing countries, can be observed [33]. This in the short run can lead to faster and cheaper trials, but then constitutes multiple ethical and other problems which had to be addressed by modifications of the Declaration of Helsinki. This issue, however, is out of the focus of this opinion statement.

Overall, the development, of which only the IIT initiation problems have been discussed here, is so discouraging for experienced physician–scientists that some have even stated “other parts of the world should learn a lesson from the misguided trial regulations that have been created in Europe” [34]. Should a modern science- and evidence-based society such as ours not be more responsive to such criticism?

## 5. Conclusions and Recommendations 

Many of the various specific steps to activate an IIT are without scientific evidence for reaching better trial quality or higher trial participant safety. There are insufficient quantitative and comparative data justifying this enormous bureaucratic effort. In Germany, the emergence of three independent but concurrent system changes, including the UH legal structure, the dual system for UH patient care reimbursement and academic research financing together with the EU CTD 2001/20 and the resulting national drug regulation (AMG, 12. Novelle), have enormously increased timelines and costs for initiation of a clinical trial, in particular an IIT. Criticism often targeting the National Competent Authorities or the Ethical Boards for this fails, as these institutions have to act as guardians of a legal system they have not created. Since the success of academic studies, e.g., the TOS, is evident, academic clinical research without primary commercial interest is of utmost importance and should be supported and protected in our society. The legal structure and financing system of university medicine have to be urgently addressed by the members of parliaments and the administrators within governmental structures on a national level with expert advice of physician–scientists concerning the problems discussed here. The European Commission with ICH and EMA should take the initiative and completely revise the GCP guidelines in a task force with active and at least equal participation of physician–scientists who are treating patients on a daily basis in clinical trials, as opposed to regulators and members of the pharmaceutical industry. Similar initiatives have been started in the US [35]. Additionally, individual versus collective ethical aspects regarding GDPR-related obstacles for clinical research should be rebalanced [36]. Finally, the members of the European parliament and the administrators within governmental structures in particular are asked to act with evidence-based and reasonable legal consequences.

## Figures and Tables

**Figure 1 cancers-14-01308-f001:**
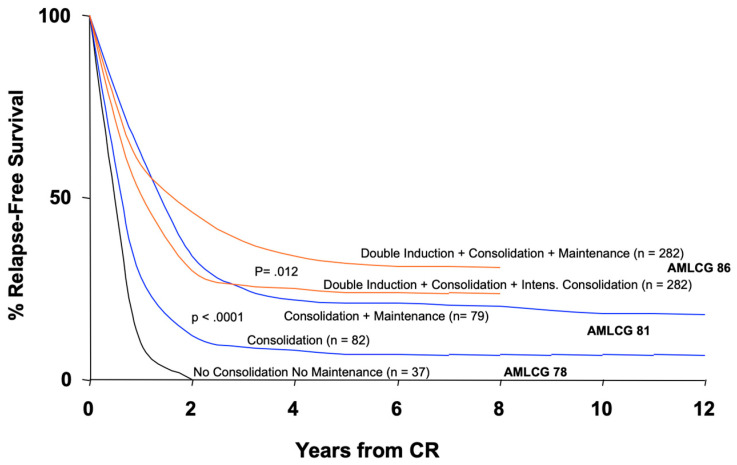
Results of AMLCG study generations 1978, 1981, 1986 for consecutive chemotherapy treatment algorithms (without bone marrow transplantation data). Patient´s age within these studies was 16–82 years. Curves of identical colours show the results of randomized comparisons of different treatment strategies. CR, complete remission (Graph kindly provided by Profs. T. Büchner and W. Hiddemann, AMLCG; reprinted in an adapted form by permission from *Nature*/*Springer*; *Current Treatment Options in Oncol.*; *Maintenance for acute myeloid leukemia revisited*, Büchner, T. et al., 2007, 8, 296–304) [7].

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
