# Peer review of "Activation of Investigator-Initiated Clinical Trials with a Pharmaceutical for Cancer Patients before and after Post-Millennial Changes of Regulations in Germany and Europe"

_cancers, 2022, doi:10.3390/cancers14051308_

Round 1
Reviewer 1 Report
1. For english editing, it appears writer might not be a native speaker, thus there needs to be an extensive check for sentence formation to make it more logical and easy to follow. Ex
Together these changes precipitated into created an (line-19).
2. The use of word author to describe various studies is confusing, at-least in the beginning there should be name ex. John et.al and then they could be referred as authors.
3. While this is an opinion, it would be great to see some data, most importantly adverse issued after drug approvals for the old and new system.
Author Response
Reviewer 1:
- …moderate English changes… AND 1. For english editing, it appears writer might not be a native speaker, thus there needs to be an extensive check for sentence formation to make it more logical and easy to follow. Ex…Together these changes precipitated into created an (line19).
Response: Two of my sons are physicians, native american english speakers and US citizens. They have checked the language and there are some changes in the revision also concerning the reviewers detailed requests (see below). Specifically, I have changed “precipitated into” into “created an” as proposed.
- The use of word author to describe various studies is confusing, at-least in the beginning there should bename John et.al and then they could be referred as authors.
Response: This suggestion has been followed where applicable.
- While this is an opinion, it would be great to see some data, most importantly adverse issued after drugapprovals for the old and new
Response: The central problem is that these comparative data – possibly justifying the bureaucratic system described – are not existent. I have mentioned this in a way hopefully better understandable in line 392 f of the revised manuscript.

Reviewer 2 Report
This is a very fine narrative story from the sight of an academic clinician with very broad experience in clinical research.
However some questions remain open and could be easily incorperated into the mansucript:
1) Are there differences of time horizonts of IITs versus sponsered studies (pharmacuetical companies and other spoinsors)
2) Are there differences... regarding costs
3) are there differences... regarding quality) . In my fiield, IITs sometimes are used to reduce cost (and burocracy and evtl. quality), while in reality fullky driven by companies.
Finally, the author implies the benefit of IIT, but he should be more cautious (e.g. in the last sentence of abstract) in this poiint. Is there a benefit for certain patients of future patients and whole society by making IITs more easy and faster. Form point of view from pharamcuetical companies is that .overregulation in Germany transfered reasearch into other countries, even outside Europe-
Author Response
Reviewer 2:
This is a very fine narrative story from the sight of an academic clinician with very broad experience in clinicalresearch.
Response: I thank the reviewer for this comment.
However some questions remain open and could be easily incorperated into the mansucript:
- Are there differences of time horizonts of IITs versus sponsered studies (pharmacuetical companies andother spoinsors)
Response: There is indeed a wide range of time spans and they are certainly overlapping also with regard to different sponsors. I have mentioned this impression for better understanding in line 269 f of the revised manuscript. Detailed comparisons, however, cannot be retrieved from the literature - unless the reviewer knows details? In this case, I would appreciate a hint where to find this for quotation.
- Are there .. regarding costs
Response: To my knowledge there is indeed no quantitative and accumulated comparison of costs between industrial and academic sponsored studies. My own experience on which I report is that the pharmaceutical industry tries to support IITs as they remain influential and this trial type is cheaper and redistributes some of the costs to the interested scientists and clinicians. However, this opinion paper rather deals with personal experience in the bureaucratic process to activate an IIT and not so much with the fact of non-existing quantitative data on this matter.
- are there .. regarding quality) . In my field, IITs sometimes are used to reduce cost (andburocracy and evtl. quality), while in reality fullky driven by companies
Response: This is exactly my impression (see also response to question 2). However, the overall judgement of trial types is complex and not in the focus of this paper. It is certainly an excellent topic for a next – more quantitative – paper.
- Finally, the author implies the benefit of IIT, but he should be more cautious (e.g. in the last sentence ofabstract) in this Is there a benefit for certain patients of future patients and whole society by makingIITs more easy and faster. Form point of view from pharamcuetical companies is that .overregulation inGermany transfered reasearch into other countries, even outside Europe-
Response: In accordance with the reviewers request, I have changed the respective parts of the manuscript to a more cautious wording and specifically introduced a balanced statement in lines 374 ff of the revised manuscript. The correct impression of the reviewer of the “international drift” of trial activities into countries outside Europe is mentioned in lines 380 ff of the revised manuscript.

Reviewer 3 Report
Dear Dr. Berdel,
thank you very much for you work, which made me feel less alone during the days lost with unnecessary bureaucracy.
I think the paper is very interesting because it deals with issues related to clinical research that often do not find space in journals, because they are not clinical or do not concern data.
In Italy we suffer a lot from this problem because our bureaucracy is so complex that it was not possible to directly implement Regulation 536 (we needed national legislative passages, not yet complete).
Below you will find some minor comments / suggestions:
- you have rightly reported the bureaucratic delays and costs. Have you calculated the deviation from the initial GANTT caused by the new "around 2000" regulation?
- it would be interesting, especially to facilitate reading, to add an image (timeline) showing the new regulatory introductions over time
-The problems you reported have been addressed by other authors. Here are some sources that I think it would be interesting to analyze and report:
- Perrone F, Marangolo M, Di Costanzo F, Colucci G, Repettos L, Merlano M, De Placido S, Torri V, Comella G, Labianca R, Parisi V, Gallo C. Cost of insurance policies for investigator-initiated cancer clinical trials in Italy. Tumori. 2005 Jul-Aug;91(4):373-9. PMID: 16277110.
- Gobbini E, Pilotto S, Pasello G, Polo V, Di Maio M, Arizio F, Galetta D, Petrillo P, Chiari R, Matocci R, Di Costanzo A, Di Stefano TS, Aglietta M, Cagnazzo C, Sperduti I, Bria E, Novello S. Effect of Contract Research Organization Bureaucracy in Clinical Trial Management: A Model From Lung Cancer. Clin Lung Cancer. 2018 Mar;19(2):191-198. doi: 10.1016/j.cllc.2017.10.012. Epub 2017 Oct 28. PMID: 29153968.
- Addressing Administrative and Regulatory Burden in Cancer Clinical Trials: Summary of a Stakeholder Survey and Workshop Hosted by the American Society of Clinical Oncology and the Association of American Cancer Institutes
Julie M. Vose, Laura A. Levit, Patricia Hurley, Carrie Lee, Michael A. Thompson, Teresa Stewart, Janie Hofacker, Suanna S. Bruinooge, and Daniel F. Hayes - De Feo G, Frontini L, Rota S, Pepe A, Signoriello S, Labianca R, Sobrero A, De Placido S, Perrone F. Time required to start multicentre clinical trials within the Italian Medicine Agency programme of support for independent research. J Med Ethics. 2015 Oct;41(10):799-803. doi: 10.1136/medethics-2012-100803. Epub 2015 Jun 11. PMID: 26066362
-I recommend creating a table that compares some major topics (eg: assurance, cost, time to study start ...) before and after the 2001 directive
-In the text, only a mention was made of the GDPR which, however, in some countries, due to national implementation and interpretation, has greatly complicated the bureaucracy. Did it happen to you too? Advice for further information Cagnazzo C. The thin border between individual and collective ethics: the downside of GDPR. Lancet Oncol. 2021 Nov;22(11):1494-1496. doi: 10.1016/S1470-2045(21)00526-X. PMID: 34735806.
- In the paper, clinicians are reported as the only "victims" of this state of affairs. Do you manage all the trials bureaucracy yourself? Do you have no support infrastructure, for example clinical research coordinator?
Finally, a major comment. A piece of history is missing: the text mentions Regulation 536/2014, which at least "on paper" should greatly streamline bureaucracy and research times and costs. What do you think? How do you think the situation can change compared to what is described?
Author Response
Reviewer 3:
… thank you very much for you work, which made me feel less alone during the days lost with unnecessarybureaucracy.
I think the paper is very interesting because it deals with issues related to clinical research that often do not findspace in journals, because they are not clinical or do not concern data.
In Italy we suffer a lot from this problem because our bureaucracy is so complex that it was not possible to directlyimplement Regulation 536 (we needed national legislative passages, not yet complete).
Response: I thank the reviewer for his kind remarks. He is - as many – obviously a victim of this bureaucratic system.
Below you will find some minor comments / suggestions:
- - you have rightly reported the bureaucratic delays and Have you calculated the deviation from theinitial GANTT caused by the new "around 2000" regulation?
Response: No, I am only reporting own experience as an example. The suggestion to gather quantitative data on this problem is very valuable. But this is an enormous effort and can only be carried out by larger institutions such as ESMO or EORTC - perhaps. We should after publication of this opinion paper undertake a new action and approach ESMO to obtain support for such an effort and I would appreciate help in this matter.
- - it would be interesting, especially to facilitate reading, to add an image (timeline) showing the newregulatory introductions over time
Response: I am at the moment only in the position to report own experience and all relevant times are mentioned in the text. Everything above this would need a larger investigation possibly on a European level (see also point 1) and thus, for the time being a figure or a table quantifying this or relating it to specific dates would be rather pseudo-exact. I prefer not to add a timeline.
- -The problems you reported have been addressed by other Here are some sources that I think itwould be interesting to analyze and report:…
Response: I thank the reviewer for these valuable suggestions and have added some of the publications directly related to oncology and the topic of the manuscript to the reference list in the revised version of the manuscript.
- -I recommend creating a table that compares some major topics (eg: assurance, cost, time to study start ...)before and after the 2001 directive
Response: Please be referred to point 2. There is variation of these data points and it would need a larger quantitative investigation to report correct data on this.
- -In the text, only a mention was made of the GDPR which, however, in some countries, due to nationalimplementation and interpretation, has greatly complicated the
Did it happen to you too? Advice for further information Cagnazzo C. The thin border between individual andcollective ethics: the downside of GDPR. Lancet Oncol. 2021 Nov;22(11):1494-1496. doi: 10.1016/S1470-2045(21)00526-X. PMID: 34735806.
Response: Thank you for this valuable suggestion. Although the GDPR consequences for science certainly need a specific investigation and publication, I have incorporated this point with the reference suggested into the revised version of the manuscript.
- - In the paper, clinicians are reported as the only "victims" of this state of Do you manage all the trialsbureaucracy yourself? Do you have no support infrastructure, for example clinical research coordinator?
Response: We had to build up this support and the resulting problem of “detouring” qualified persons from the clinical science and patient “front” to the “rear”, which dramatically adds to our staff shortage has been mentioned in the manuscript (line 335 ff).
- Finally, a major A piece of history is missing: the text mentions Regulation 536/2014, which atleast "on paper" should greatly streamline bureaucracy and research times and costs. What do you think?How do you think the situation can change compared to what is described?
Response: I am afraid that in stark contrast to the intention of EU commissioner John Dally („I want an assessment system that is fast, slim, pragmatic, and not disproportionately expensive, complex, or bureaucratic.“) EU CTR 536/2014 will blow up bureaucracy to another quality level. Just take the 7 years it needed to install a still non-functional CTIS system… It is, however, fair to collect experience on all levels first and then report on the experiences with this new regulation. The CTR 536/2014 is mentioned in lines 326 ff.
